# Urban Forest Recreation and Its Possible Role throughout the COVID-19 Pandemic

**Dastan Bamwesigye** [1,2], **Jitka Fialova** [1,*], **Petr Kupec** [1], **Evans Yeboah** [2,3], **Jan Łukaszkiewicz** [4,*], **Beata Fortuna-Antoszkiewicz** [4] **and Jakub Botwina** [4]

1    Department of Landscape Management, Faculty of Forestry and Wood Technology, Mendel University in Brno, Zemědělská 1, 613 00 Brno, Czech Republic; dastan.bamwesigye@mendelu.cz (D.B.); petr.kupec@mendelu.cz (P.K.)

2    Department of Forest and Wood Products Economics and Policy, Faculty of Forestry and Wood Technology, Mendel University in Brno, Zemědělská 1, 613 00 Brno, Czech Republic; xyeboah1@node.mendelu.cz

3    Department of Business Economics, Faculty of Business and Economics, Mendel University in Brno, Zemědělská 1, 613 00 Brno, Czech Republic

4    Department of Landscape Architecture, Institute of Environmental Engineering, Warsaw University of Life Sciences—SGGW, ul. Nowoursynowska 159, 02-776 Warszaw, Poland; beata_fortuna_antoszkiewicz@sggw.edu.pl (B.F.-A.); jakub_botwina@sggw.edu.pl (J.B.)

\*    Correspondence: jitka.fialova@mendelu.cz (J.F.); jan_lukaszkiewicz@sggw.edu.pl (J.Ł.)

**Abstract:** Urban forests provide benefits in terms of the environment and society. Many people living in urban areas could profit from the recreational services provided by urban forests to alleviate the physical and psychological stresses caused by closure restrictions during the COVID-19 epidemic. However, limited research has been conducted on the role of forest recreation during the COVID-19 lockdown to support future policy decisions regarding such dramatic circumstances. The study aims to investigate the frequency of visitors in the reference Training Forest Enterprise Masaryk Forest Křtiny in Brno, Czech Republic, and to verify if the lockdown led to its intensified use, under the context of weather conditions. Data were collected using a Pyro Box Compact reader and TRAFx Infrared Trail Counter. Regression analysis was performed to determine whether individual weather factors significantly influenced the intensity of forest visitations before and during COVID-19. The number of visits in 2021 during the strict lockdown and post-lockdown periods was significantly higher during spring and autumn than in 2016 and 2017, respectively. The frequency pattern of forest users visibly changed in 2021, and the total number of visits was much higher, even though 2021 had worse weather than the reference years before the pandemic. The results show the recreational use and importance of the forest, especially during the pandemic. This work is a continuation of our previous research with regard to this area, which indicated the importance of forest recreational services for the well-being and health of city dwellers.

**Keywords:** COVID-19 pandemic; municipal forests; forest recreational services; weather conditions

## 1. Introduction

The powerful influence of vegetation and green spaces on human quality of life is undisputed, with aesthetic, emotional and physiological benefits [1–4]. In Europe, since the 19th century, lush urban greenery has been considered by city dwellers as a factor that has a relaxing, calming and therapeutic effect on the human body, both physically and mentally. The favourable effects of green areas on the human body are well documented and include the following: reduced stress levels during leisure time, the improvement of social interaction in public spaces, the acceleration of recovery from illness (e.g., green areas around hospitals and mental health institutions), the reduction of mental fatigue, the improvement of concentration and performance (e.g., greenery around schools, kindergartens, workplaces) and the suppression of feelings of aggression and violence. Research into

the health-promoting effects of vegetation on the human body has been put into practice for years, for example, in Japan, as part of therapies known as "forest bathing" (Japanese: "shinrin-yoku") [5,6]. Much evidence shows that contact with nature—especially lush vegetation—helps to reduce stress hormones in the human body and stimulates the activity of white blood cells responsible for fighting cancer cells and infections [3,7–9].

Recent observations confirm the well-known fact of the extensive and beneficial effects of urban green spaces (UGS), e.g., urban parks, urban forests and woodlands, on human health (emotional and physical) [10–14]. Contact with nature, predominantly vegetation, can also prevent or significantly reduce human mental and physical health risk factors. Natural areas are vital in cities, where most of the world's population (expected to reach 5 billion by 2030) is already living [15]. The highly positive role of UGS—e.g., urban forests—has already been demonstrated in the context of mitigating the effects of the global SARS-CoV-2 (COVID-19) pandemic (the outbreak of which was declared by the World Health Organization on 11 March 2020 [16]).

Over the past two years, the COVID-19 pandemic has led to unprecedented changes in the daily lifestyles of most people around the world. The imposed public health and economic restrictions of social distancing, lockdowns, domestic isolation and drastic reduction of time spent outdoors have left a strong mark on the mental and physical state of millions of people [6,10–14,17]. The long-term stress of lockdowns was found to affect not only the human mind but also immunity against diseases and the overall activity of the body on a biological level [5]; this was reported in the context of the global community by the WHO well over a decade before the outbreak of the pandemic [18]. The COVID-19 pandemic in 2020–2021 changed the lifestyles of local communities and entire societies [11,19–33]. Globally, pre-pandemic UGS, such as parks or primeval forests, provided many services and benefits to people and biodiversity. In many countries, parks and urban forests have experienced dramatic increases in recreational intensity. The catalyst is the human need for contact with nature and the desire to maintain social ties despite localization. Particularly in urban forests—located on the outskirts of cities—significant increases in visitation have been observed, e.g., in Bonn or Freiburg (Germany) [11,20], Krtiny (Czech Republic) [19], Burlington, Vermont (USA) [26] and Kanas (China) [21].

Urban forests, national parks and natural recreational forests are places where stress can be reduced. Studies on the effects of the forest on satisfaction, relaxation, quality of rest and recovery from stress have been conducted in many places during the pandemic, such as Kanas Forest (China) [21]. Online surveys have often been used as an instrument to evaluate satisfaction with being in nature, using tools such as the Perceived Restorativeness Scale (PRS) and the Psychosocial Well-being Index Short Form (PWI-SF) [34]. Such social surveys show the growing need for and value of such areas in times of crisis, such as COVID-19. In Burlington, Vermont (USA), for example, people reported that these areas were important for a wide range of activities, from exercise to bird watching, but also for stress reduction in times of global chaos [25]. Similarly, a comparative online survey conducted during the pandemic (March–May 2020) in Croatia, Israel, Italy, Lithuania, Slovenia and Spain found that urban residents tended to need accessible green areas in response to the pandemic, mainly for exercise, relaxation and nature observation [28].

Limited research has focused on the effect of the COVID-19 epidemic on the increased movement of people in the forest environment and the exploitation of the positive effects of the forest on human health (e.g., [6,22,34–41]). Moreover, there has been little research (e.g., [42]) on the effect of restrictions associated with the reduction in movement of people during the epidemic, including in the forest environment. While the changes in the models for leisure in the open air were noticeable during the COVID-19 pandemic, some researchers point out that there is still little research conducted considering the multi-seasonal aspect that would allow a fair comparison of recreation visitation data in a given location, e.g., a forest with variable weather conditions [23,43].

The situation in the Czech Republic in March and April 2021 was challenging. A hard lockdown was announced on 26 February 2021, at a press conference following a

Cabinet meeting. The government announced restrictions on movement within the district of residence. The police and the army checked travel permits. The government revoked a regulation that allowed wearing two surgical drapes instead of an FFP2 or KN95 respirator or a nano-muff. On 22 March 2021, the lockdown announced by the government on 1 March was extended. However, playing sports or walking around the district was now possible. On 12 April 2021, some of the measures were relaxed. Zoos and botanical gardens were allowed to open outdoors. At the same time, the state of emergency ended with a ban on travel between districts and a curfew between 9 p.m. and 5 a.m. Urban forests, located close to cities, are essential for the landscape's biodiversity and extremely important for the urban environment. The presented study examined the frequency of forest visitors in the Training Forest Enterprise Masaryk Forest Křtiny (TFE MF Křtiny) near the City of Brno. It aims to compare the recreational traffic intensity in the forest during the COVID-19 pandemic to the preceding period to verify the above thesis that the pandemic led to its intensification. Concerning the current state of knowledge, we expect that our research could help clarify the general phenomena related to the recreational use of the urban forest in the context of the pandemic and lockdown in the Czech Republic and globally, showing interesting patterns of people's monthly attendance in the forest, referring to typical conditions. In addition, our research highlights an attempt to determine the time of year when recreational attendance in the forest was mainly dependent on the impact of the pandemic and lockdown—when people visited the forest more frequently compared to reference years. Moreover, this study aims to provide policy and decision makers in future situations similar to COVID-19 with information for decision making as regards the physical and psychological welfare of urban citizens using forest recreational services.

## 2. Materials and Methods

### 2.1. Study Site

The research site was the Training Forest Enterprise Masaryk Forest Křtiny [44]. It is an integral part of the Mendel University in Brno (Czech Republic) and is a special-purpose institution of the same university's Faculty of Forestry and Wood Technology. The forest property covers an area of 10,265 ha. The majority of the stand is characterised by a natural species composition, which has remained unaltered over time. Moreover, the forest service intentionally maintains and extends the forest's local natural values. The primary indigenous woodland species are spruce and pine as conifers and larch, beech and oak as broadleaves. As a result of the successful reintroduction of some native tree species and the effective attempt to transform the stand towards a selective forest, TFE MF Křtiny presents an exceptional combination of natural, landscape and educational elements.

The afforested areas of the TFE MF Křtiny form a continuous complex immediately adjacent to the northern borders of the City of Brno (Figure 1) and extend as far as the city of Blansko. After thorough consultations with foresters, we chose this location as representative for location of the counter (one important entrance). The area is relatively close to three car parks (one located directly on the border of the City of Brno), and the bus stop is 1.5 km from where the counter is located. The opposite direction from which visitors can reach this area is by taking the trains Brno-Bílovice and Svitavou and then walking back to the City of Brno, passing by the counter.

### 2.2. Stages of Research and Methods Used

The content of the research was defined by the research questions (addressed to the relevant literature), which were framed as follows:

- How did municipal forests affect social health during the COVID-19 pandemic?
- Do weather conditions influence forest recreation activities in the Czech Republic; did they determine the quantity of forest recreation during the pandemic?

Based on the answers to the above questions, the authors then formulated the following research questions (relating to the data collected during the fieldwork and observations):

- What was the relation between the number of visitors to TFE MF Křtiny and primary weather conditions (air temperature and precipitation) observed for the selected data?
- Was there a visible change in the public need for recreation in TFE MF Křtiny when comparing the years prior to and during the COVID-19 pandemic?

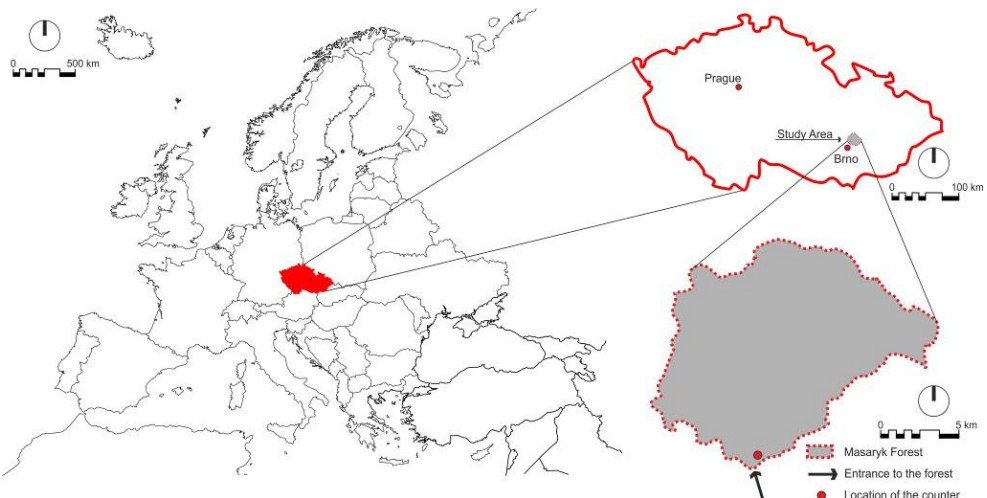

**Figure 1.** Moravian metropolis (the City of Brno) and the location of TFE MF Křtiny (elaborated by authors).

Regarding the aim of the study, extensive research of the related literature was conducted. We used qualitative approaches in accordance with Creswell [45], such as case studies, exploration and description. Therefore, we used exploratory methods based on the TFE MF Křtiny case study and content analysis of the literature on the subject.

Some actual databases, such as Scopus, Web of Science and Google Scholar, were searched for related study materials. Primary keywords included were the COVID-19 pandemic, lockdown, municipal forests, forest recreational services, urban green areas, weather conditions, and thermal comfort. We also reviewed local authorities' resolutions and regulations concerning public health in the Czech Republic during the lockdown. Based on all the materials, TFE MF Křtiny became, in our study, the representative case of a municipal forest exposed to increasing recreational movement during the COVID-19 pandemic.

The study applied quantitative approaches to collect a climatic dataset of local basic climate conditions (precipitation and air temperature-daily) using a dataset collected from the official climatic station in Brno-Tuřany for the years 2016, 2017 and 2021 (official climatic station of Czech Hydrometeorological Institute—location: Tuřany Airport (TA)) and from the unofficial climatic station owned by the Mendel University in Brno for the years 2016, 2017, 2021 and 2022 (data series collected every 15 min). As the official data for 2022 were unavailable, the authors sought a correlation between both datasets to use unofficial data for the whole period. Correlation coefficients using the observations 1%–1161.5% critical value (two-tailed) = 0.0575 for $n$ = 1161. Climate data validation was done to justify using the climatic data from TFE MF Křtiny.

The main objective of monitoring forest visitors' activities is to provide basic information on the number of visitors and data on the temporal variability of traffic. We placed an automatic counter for the number of pedestrians and cyclists to monitor a selected representative forest road in the TFE MF Křtiny (Figures 2–4). The monitoring equipment was installed in July 2014, and the monitoring took place (with some interruptions) until September 2022. An Eco-counter automatic Pyro Box Compact reader was used to monitor trail users from 2014 to 2018. This device counts all users moving along the main forest trail leading from the city of Brno without distinguishing between them (hikers, bikers, in-line skaters and others). The temperature difference between the human body and its surroundings is the basis of the measurement technology [46].

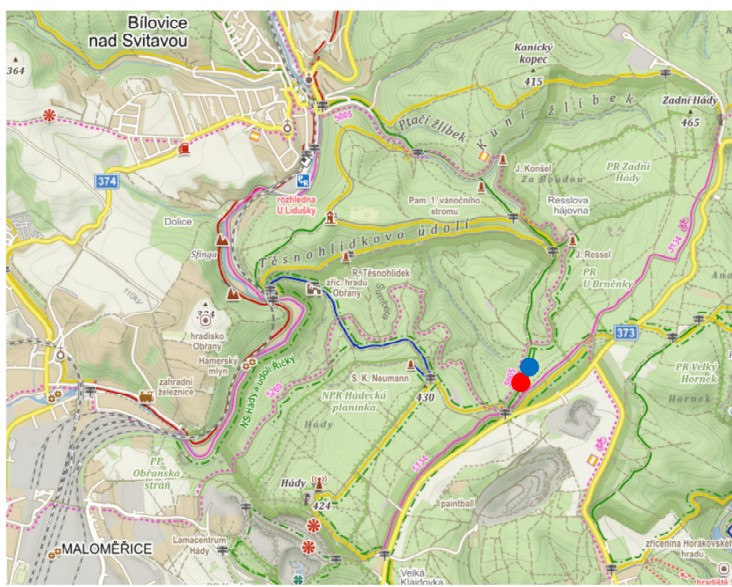

**Figure 2.** Evaluated part of TFE MF Křtiny (blue and red dots show where the counters were installed) [47].

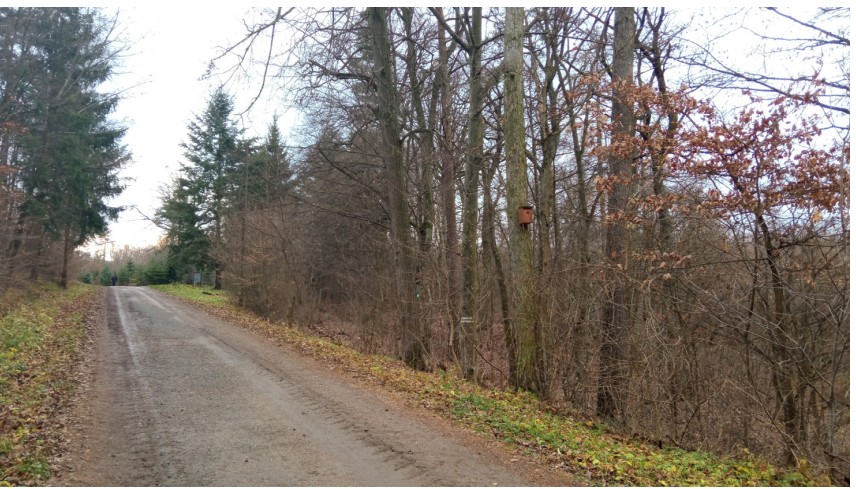

**Figure 3.** Automatic counter hidden in the bird box (photo by: Fialová, 2022).

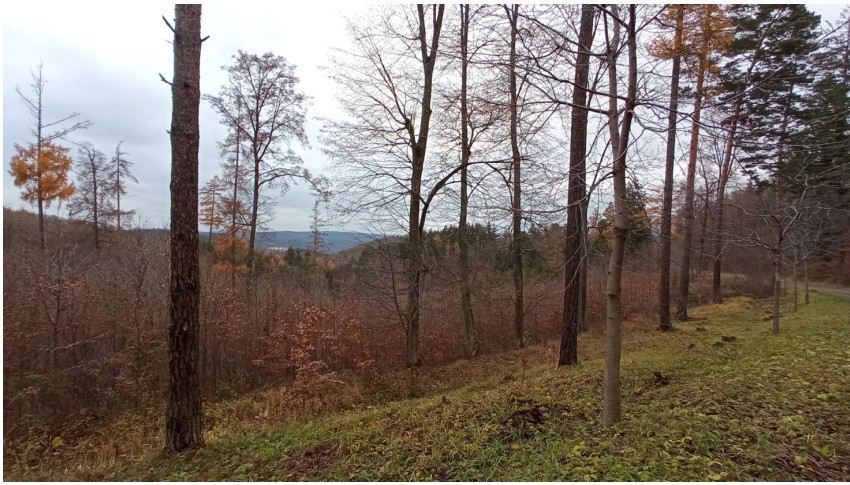

**Figure 4.** Forest stand and the view from the location where counter is installed (photo by: Fialová, 2022).

In 2021, a TRAFx system—an infrared distance counter—was installed. Its advanced microelectronic technology and high-quality infrared detector allow for measuring all infrared radiation emitted by passing people. According to [46], this technology works well for counting people moving along the forest trail. Finally, it has a very compact, discrete design that minimises the chances of vandalism. In winter conditions, on trails used for snowshoeing, skiing or snowmobiling, the TRAFx infrared trail counter works well. The use of automatic counters made it possible to determine the daily number of forest users during the studied years. Both types of counters provide total passes at hourly intervals.

We analysed a vast dataset collected over years in TFE MF Křtiny. The numbers of visitors entering the forest before (2016 and 2017) and during the lockdown year (2021) were arranged into one database [48]. Next, for comparative purposes, we introduced a specific range of months (March–December) in each year covered by the study (2016, 2017 and 2021). We also used visitor data from a specific month interval for post-lockdown year 2022 (March–September) for comparison. The statistical analysis was performed with the Statistica 13.0 software.

First, we compared month-by-month numbers of forest visitors in the reference years (2016 and 2017) and lockdown year (2021). Month-by-month differences in visitor numbers between 2021 vs. 2016 and 2021 vs. 2017 were checked. The next step focused on regression tests concerning the increased frequency of forest visitors between 2016–2017 and 2021. Data from two reference years (2016 and 2017) and the lockdown year (2021) were used to compare their means and standard errors. The results were configured into pairs containing the reference years (2016 and 2017) and lockdown year (2021) and independently obtained groups of results to determine if the lockdown period affected visiting frequencies. For this purpose, a simple *t*-test was applied, as the groups had their distributions close to normal and sufficient levels.

Finally, the relationship between weather conditions and forest visit numbers from March to December during reference years (2016 and 2017) and the lockdown year (2021) was analysed. The regression analysis was performed to determine whether individual weather factors (temperature and precipitation) significantly influenced the intensity of forest visitations.

The year 2018 was omitted from the analysis because of missing data; information was available only up until May. We omitted this year because we aimed to investigate whether forest visitation was influenced by the weather or the COVID-19 pandemic.

## 3. Results

### 3.1. Climatic Dataset Analysis

The data from both sites of Tuřany Airport and TFE MF Křtiny were found to have similar values with strong positive correlation (Tables 1 and 2).

**Table 1.** Correlation between precipitation (mm) of Tuřany Airport and TFE MF Křtiny.

| Precipitation TA | Precipitation TFE | |
|---|---|---|
| 1.0000 | 0.6096 | Precipitation TA * |
| - | 1.0000 | Precipitation TFE ** |

* TA—the climatic station of Czech Hydrometeorological Institute, Tuřany Airport, Brno; ** TFE—TFE MF Křtiny.

**Table 2.** Correlation between air temperature (°C) of Tuřany Airport and TFE MF Křtiny.

| Temperature TA | Temperature TFE | |
|---|---|---|
| 1.0000 | 0.9730 | Temperature TA * |
| - | 1.0000 | Temperature TFE ** |

* TA—the climatic station of Czech Hydrometeorological Institute, Tuřany Airport, Brno; ** TFE—TFE MF Křtiny.

Observations of TFE MF Křtiny forest visits in 2016, 2017 and 2021, conducted from March to September, and in 2022 from January to September, were further investigated. During the observation periods, weather conditions were found to have a significant but similar effect on visit frequency in the years studied (Table 3). The regression analysis of the effects of the individual factors (temperature and precipitation) showed significant influences on the intensity of forest visitation. The multiple regression of these factors showed a certain degree of synergy (Table 4). This is evident when looking at the coefficients of determination ($r^2$, which measures the goodness of fit). Interestingly, in the reference years (2016 and 2017), the $r^2$ values, which also explain the proportion of the total influence (value of 1), are not significantly different.

**Table 3.** Weather conditions and forest visit numbers during reference years (2016 and 2017) and lockdown year (2021) from March to December.

| Year (March–December) | Annual Mean Air Temperature (°C) | Precipitation—Total Annual Sum (mm) | Total Visit Number |
|---|---|---|---|
| 2016 | 11.5 | 580.9 | 62,497 |
| 2017 | 11.6 | 552.7 | 62,039 |
| 2021 | 10.9 | 629.3 | 66,583 |

**Table 4.** The effect of weather factors on the number of forest visitors (y) during a given year.

| Year | Single Factors | | Combined Factors | Regression Formula for Combined Factors |
|---|---|---|---|---|
| | Temperature | Precipitation | Temperature and Precipitation | |
| 2016 | r = 0.4334 $r^2$ = 0.1878 $p < 0.001$ | r = −0.1907 $r^2$ = 0.0360 $p = 0.0015$ | r = 0.4940 $r^2$ = 0.2441 $p < 0.001$ | y = temp × 89,967 + prec × (−61,120) + 1,382,041 |
| 2017 | r = 0.3811 $r^2$ = 0.1453 $p < 0.001$ | r = −0.2673 $r^2$ = 0.0700 $p < 0.001$ | r = 0.4778 $r^2$ = 0.2226 $p < 0.001$ | y = temp × 81,096 + prec × (−78,209) + 14,218,671 |
| 2021 | r = 0.3095 $r^2$ = 0.0958 $p < 0.001$ | r = −0.2441 $r^2$ = 0.0714 $p < 0.001$ | r = 0.4352 $r^2$ = 0.1894 $p < 0.001$ | y = temp × 77,393 + prec × (−71,339) + 1,582,030 |

where: r—the regression coefficient; $r^2$—determination coefficient; *p*—coefficient of significance for the regression formulae. Obtained determination coefficients and low "*p*" values point to a significant (circa 20%) participation of weather conditions in the frequency of visiting of TFE MF Křtiny.

### 3.2. Visitor Dataset Analysis

The reference years (2016 and 2017)—before the COVID-19 pandemic—were characterized in TFE MF Křtiny by very similar monthly forest visits. The first peak in May reached about 10,000 in 2016 and 2017 (Figure 5). A decline followed this during the summer holidays. Then, a second significant increase—albeit at a lower level—occurred in August (8000–9000). The third and weakest quantitative jump in attendance occurred in October, reaching approximately 5000 entries. The lowest number of visits occurred at the end of September and the beginning of October.

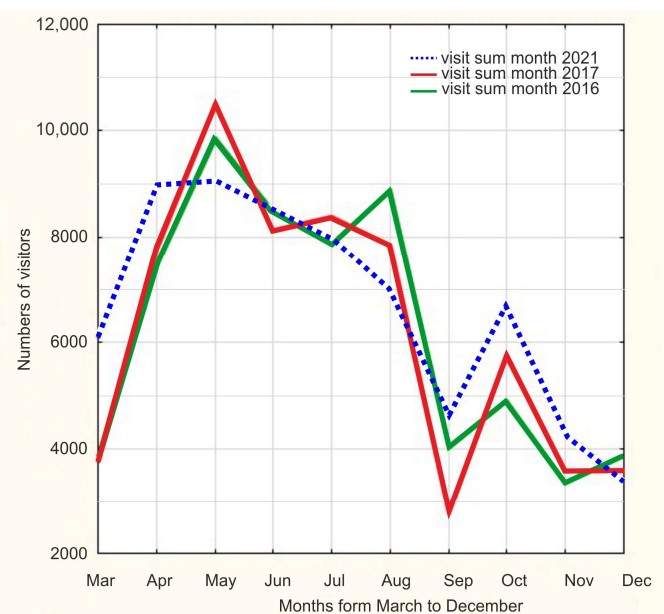

**Figure 5.** Month-by-month numbers of forest visitors in reference years 2016 and 2017 and lockdown year 2021 (Statistica 13.0 software).

Attendance in the lockdown year (2021) was different. The number of visits to the TFE MF Křtiny forest in March and April was higher than in the corresponding months of the reference years. In May, the number of visits was high (about 9000) but lower than in the reference years (Figure 5). During the summer holidays, the number of visits to the forest was similar between the reference years (2016 and 2017) and the lockdown year (2021), decreasing until September and showing a similar trend. However, from September to December 2021, there was an apparent increase in forest visits. The most significant differences from the reference years were observed at the end of September and the beginning of October, when the number of visits to the forest was almost 2000 higher than the reference years (Figure 6). The graph shows month-by-month readouts of user frequency in the forest, indicating clear differences in visitor numbers between 2021 and 2016 (green) and between 2021 and 2017 (red).

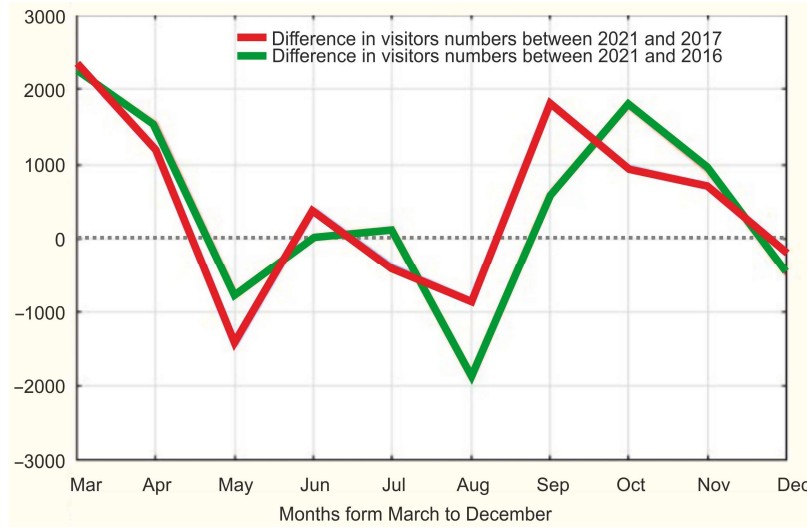

**Figure 6.** Month-by-month visitor number difference between 2021 and 2016 (green) and between 2021 and 2017 (red). Readings above the "zero line" indicate a higher attendance in 2021 (Statistica 13.0 software).

Despite the differences in the quantitative comparison of monthly visits with the reference years, 2021 is characterized by a much higher total number of forest visits (>4 thousand). The difference is even more significant when weather conditions are taken into account. The year 2021 was characterized by a lower mean annual air temperature and a higher total precipitation than 2016 and 2017 (Table 3).

The results obtained at this stage showed a clear advantage in the number of TFE MF Křtiny visitors in 2021 compared to the reference years 2016 and 2017 in spring (March–May) and autumn (September–November). In the presented study, the dependent variable is the attendance of users in the forest, with time (season of the year) and weather conditions as independent variables.

The next step was to determine whether the COVID-19 pandemic had a significant effect on the frequency of visits; the results were configured into pairs with lockdown year (2021) and reference years (2016 and 2017) as independently obtained groups. A separate study was conducted for independent trials in two replicates: 2021 vs. 2016 and 2021 vs. 2017—each divided into spring and autumn periods.

Data from the lockdown year (2021) and reference years (2016 and 2017) were used to compare means and standard errors. A simple t-test was used, as the groups had close to normal distributions and sufficient numbers. The analysis results are presented in the tables (spring and autumn). In all cases, the mean number of visitors/day is visibly higher in 2021, regardless of the season and the reference years (2016 and 2017). Nevertheless, the marginal significance was only observed in 2021/2016 spring and the 2021/2017 autumn (Tables 5 and 6).

**Table 5.** Comparison of the mean daily visiting frequencies in the spring season (March, April, May) between 2021, 2016 and 2017.

| Parameter | 2016 (Reference) | 2017 (Reference) | 2021 (Lockdown) |
|---|---|---|---|
| Mean: Visitors/day | 243 | 261 | 300 |
| *s* (std dev.) | 195.72 | 195.91 | 182.25 |
| *n* | 92 | 92 | 81 |
| *p* | 0.0504 | 0.1823 | - |

**Table 6.** Comparison of the mean daily visiting frequencies in the autumn season (September, October, November) between 2021, 2016 and 2017.

| Parameter | 2016 (Reference) | 2017 (Reference) | 2021 (Lockdown) |
|---|---|---|---|
| Mean: Visitors/day | 166 | 152 | 181 |
| *s* | 113.4 | 104.9 | 126.3 |
| *n* | 122 | 122 | 122 |
| *p* | 0.3267 | 0.0463 | - |

where: mean—mean number of daily visitors in each year; *s*—standard deviation; *n*—number of days taken into account; *p*—coefficient of significance between values of variables: $p < 0.05$ = difference between variables is statistically significant/$p > 0.05$ = difference between variables is not statistically significant.

## 4. Discussion and Conclusions

### 4.1. The Climate and Its Impact on Recreation in the Forest

Tourist behaviour is also significantly affected by climate factors and global warming (Huisun National Forest Park (HNFP) in Taiwan). The findings show that when only changes in temperature and precipitation are considered, climate change may not reduce the total number of visitors to the HNFP [31]. In the context of this publication, intriguing results have been presented for the example of the Florida National Scenic Trail (FNST) [33].

Over five years of daily use by outdoor recreationists, the data collected using mechanical infrared counter devices indicated that days with maximum temperatures of 16–22 °C attracted the most visitors, suggesting that this is the most favourable range of daily maximum temperatures for recreation at the FNST. Daily maximum temperatures below 6 °C and above 31 °C and heat index values above 38 °C brought significantly fewer visitors. Days with heavy rainfall (>2.54 cm) or high heat index ≥ 35 °C may have negatively affected recreation participation not only on the same day but also on the next typical weather day [34]. People visited the TFE MF Křtiny more frequently despite 2021 (lockdown year) being colder and wetter than 2016 and 2017, respectively.

Finally, several global studies point to the impact of climate change on outdoor recreation and management (e.g., [31,32,49,50]). Climate change will affect outdoor tourism by changing weather patterns and affecting the supply of outdoor recreation resources [43]. Ongoing research examines whether climate change adaptation strategies utilised by parks and forest management will positively affect tourist intentions [33]. Knowing how visitors react to environmental changes is crucial to maintaining "weather resilient" visitation flows. By anticipating local disruptions and future shifts in demand, regional networking in nature-based destinations can mitigate tourism declines and adapt to changing visitor flows [49].

The so-called "recreational bioclimate" includes all biological variables that affect the human body, especially in the so-called "recreational layer"—the air zone of about 2.0 m above ground level. Apart from the topography or open surface waters, the primary influence is the vegetation, especially tree stands (e.g., in the areas of municipal forests). The influence of different landscape elements can change the parameters of the recreational bioclimate.

Tree cover can significantly influence the recreational bioclimate within the forest and the surrounding area [50–54]. Forest tree stands—if valuable for recreational purposes—should provide an appropriate spatial structure that ensures, among other things, the improvement of aeration, solar radiation, and thermal conditions [1,7,51,52,54,55]. The last of these factors is particularly important for recreation, including in the case of forests. During the day, the environmental conditions in the forest determine the body's thermal balance, translating into the potential feeling of thermal comfort. It ensures well-being and the ability to remain active for a long time. The conditions of thermal comfort during a summer day for a lightly clothed person doing little physical activity (e.g., walking, cycling or hiking) are as follows:

- a daily average air temperature of ca. 20 °C;
- a daily average relative air humidity reaching ca. 60%;
- a daily average air movement not exceeding 10.0 km/h.

It is also known that the human body's perception of thermal comfort can be found in the broader range of air temperatures from 17.2 °C to 21.7 °C, but with the distinction that the higher the air temperature, the lower the humidity should be. When the air temperature exceeds 22.0 °C, the expected average relative humidity should not exceed 20% [7,56].

Liu et al.'s [57] study investigated whether climate change adaptation strategies positively impact tourists' travel intentions. The research found that, except for heatwaves, climate change–related events such as more powerful typhoons, heavy rainfall, and cold weather would reduce travel intentions, especially for tourists with low place attachment. It is observed that maximum daily temperatures under 6 °C and over 31 °C and heat index scores over 38 °C significantly reduce recreational activity and can be interpreted as local temperature thresholds in a recreational context [58]. The season-adjusted autoregressive integrated moving mean model showed a significant negative effect of temperature, relative humidity and cold weather and a significant positive effect of weekends and holidays on leisure visits to this route. Days with high precipitation (>2.54 cm) or high heat index (≥35 °C) may have harmed recreational participation not only on the same day but also on the next typical weather day. The findings suggest that facility operators who need staff and other inputs should be prepared for a reduction in visitor numbers following bad weather, even on days representative of a typical weather day.

### 4.2. Worldwide Tourism in the Municipal Forests in the Context of the COVID-19 Pandemic

Contact with nature, mainly vegetation, can prevent or significantly reduce human mental and physical health risk factors. It is crucial in cities, where most of the world's population lives (projected to be 5 billion in 2030) [15]. The highly positive role of urban green spaces (UGS)—such as urban forests—has already been demonstrated in the context of mitigating the global SARS-CoV-2 (COVID-19) pandemic (the outbreak of which was announced by the World Health Organization on 11 March 2020 [16]). Over the past two years, the COVID-19 pandemic has led to unprecedented changes in the everyday lifestyles of most people worldwide. Imposed public health and economic restrictions in the form of social distancing, lockdowns, home isolation and drastically reduced time spent outdoors have left a strong mark on the mental and physical state of millions of people [6,10–14,17]. The influence of restrictions was severe in the area of TFE MF Křtiny, but not in the way that the authorities expected. March and April 2021 were the months with the strongest restrictions (people were not allowed to leave the district), but the number of visits was the highest in this part of the year.

The COVID-19 pandemic in 2020–2021 changed the lifestyles of local communities and entire societies [11,19–33]. Globally, pre-pandemic UGS, such as parks or primeval forests, provide many services and benefits to people and biodiversity. Particularly in urban forests—located on the outskirts of cities—significant increases in visitation were observed during this period, e.g., in Bonn or Freiburg (Germany) [11,20], Krtiny (Czech Republic) [19], Burlington, Vermont (USA) [26] and Kanas (China) [21]. The analysis of the number of visits during the strongest restrictions shows a large number of people visiting such areas.

The increase in visitation to urban forests coincides with some reports of declining use of urban parks during the pandemic. Differences were found in perceptions of green spaces' inconvenience and management needs. In Croatia, in particular, data on the use and perceptions of green spaces collected during Europe's first closure period show that people had primarily positive perceptions of green spaces, regardless of settlement size [29].

Concerning the COVID-19 pandemic, tourist traffic observations and studies are being carried out in many places worldwide, e.g., in urban forests and open areas around cities. For example, in North Carolina (USA) [24], 56% of respondents reported stopping or reducing their visits to parks, and geo-tracked park visits fell by 15%. This may have been due to increased recreational traffic and the selection of the most popular tourist sites (e.g., [19,22,24,29,30]). The increased intensity of recreation in urban forests and other open spaces during the COVID-19 pandemic requires changes in their management (e.g., [24,26,27,30–33]). The visitors' analysis results show the changes in restrictions, laws and movement control. Following Ciesielski et al. [42], the effects of the restriction policy on the number of forest visits were evident in the presented studies. Different levels of restrictions characterised the different phases of the lockdown. Public perceptions of the pandemic varied. In the initial phase, a significant percentage of the public feared the coronavirus, spurred by a fear of the unknown. This was not the case for the southern part of the TFE MF Křtiny.

For example, 14 popular European national and nature parks faced new challenges due to the pandemic in organizing tourism and recreational use. Challenges included overcrowding, a changing visitor profile, problematic attitudes and tensions among various visitor groups [26]. To contradict such phenomena, the authors recommend incorporating information campaigns, traffic management and introducing one-way systems on trails, promoting sustainable tourism models. Also, the importance of using widely available solutions—smartphone apps and information technology—could possibly enrich and facilitate the cultural aspects of forest visiting, focusing on a healthy environment and lifestyles that promote renewable energy [27].

Although changes in outdoor recreation routines were noticeable during the COVID-19 pandemic, some researchers point out that there are still few studies that consider seasonal variation, allowing a reliable comparison of recreational visitation data at a given location,

such as a forest, under changing weather conditions [23,43]. Studies that have presented visitor demand in terms of projected transformations in climate and weather are limited to predicting the demand for visitation in a single season at a single location. Reports from the Swiss National Park, for example, show that the "month criterion" was the most significant factor in predicting visits, which was followed by the exchange value, temperature, precipitation, public holidays and media coverage [32]. In this study, we proved that the weather is not a significant factor in the decision to go to the forest for a walk.

### 4.3. The Fluctuation of Tourism in the Municipal Forests Regarding Environmental Conditions

Woodland ecosystems offer many services and benefits for people and biodiversity [19]. In the aftermath of the COVID-19 epidemic, many urban residents took advantage of the recreational services of forests and parks to relieve the psychological stress caused by the closure regulations. Similarly, urban forest services provide recreational and leisure space, mental health relief and contemplation for urban residents.

According to Derks et al. [11], new census data have completed quantitative and qualitative records of municipal woodland visits, supported by interviews with many experts [11]. The number of visitors has more than doubled since COVID-19 activities began in March 2020. The visitor flows have changed dramatically, from an even spread during the day, with minor peaks before and after opening hours, to a peak in the later hours of the afternoon. Results from automatic counters show that in TFE MF Křtiny, the number of visits almost doubled in March 2021 compared to 2016 and 2017. March and April 2021 were the months with the highest restrictions but exhibited a very high number of forest users. Weinbrenner et al. [20] conducted a mixed-methods study of forests around the southern German city of Freiburg to examine how these appropriation practices worked and shed light on the importance of forests to urban residents under these circumstances.

Rogowski [24] states that variability and seasonality are crucial to the analysis of visitor numbers in the mountain national parks. A high concentration of tourists in a given location and a steady increase in the number of visitors will lead to the Tourism Carrying Capacity (TCC) of routes being exceeded and the associated problem of overtourism. In order to continue research in this area and discover methods to counteract overtourism, the number of visitors in national parks should be monitored. Two indicators were set to achieve this goal: the Visitor Index (VI) and Seasonality Index (SI). The COVID-19 pandemic remarkably altered the temporal distribution of tourist flows. Relevant data were collected daily for four years (2017–2019, with 1095 visitors per day, and pre-processed data during the COVID-19 period in 2020) from the Tourism Tracking System (MSTT). The study identified four tourist seasons: high season in summer, medium season in autumn, medium season in spring and low season in late autumn and winter. The index of the seasonality of the periods and partial periods presented indicates some variation in the activity of weekend tourism. The interpretation of the indicators allowed a precise description of the visitor flows to SMNP. The study was, therefore, able to determine the annual or monthly fluctuations in visitor flows and to identify seasons and sub-seasons based on the available daily data.

Results from Grima et al. [28] show that 69.0% of participants had more or much more frequent visits to outdoor spaces and urban forests, and 80.6% felt that the significance of and access to these areas had become more or much more important. In addition, 25.8% of the respondents had never or very rarely visited their native natural areas before the pandemic, but 69.2% of first time or occasional responders said that access to these areas was "essential" during COVID-19. This finding is significant and interesting. Research based on questionnaires is proposed to be done in the area of interest in the future.

### 4.4. The Recreation Frequency in TFE MF Křtiny during the COVID-19 Pandemic

The study aimed to determine the variations in forest recreational services in the urban forest of TFE MF Křtiny during COVID-19 and if weather conditions influenced such variations. The study assessed the frequency of forest visitors. In this study, the trend of

recreational use due to the lockdown and the COVID-19 pandemic was intensified, similar to the data reported worldwide by the cited literature [11,12].

Our results showed an increase in the number of TFE MF Křtiny visitors in 2021 compared to the reference years 2016 and 2017 in spring (March–May) and autumn (September–November). Once again, 2018 data were only available for March, April and May, and therefore could not be used for the regression analysis.

Concerning whether the COVID-19 pandemic significantly affected the frequency of visits, the summary statistics did not show a significant inhibiting influence of the weather conditions on the great social need for recreation in the forest. People visited the forest more frequently despite 2021 (lockdown year) being colder and wetter than 2016 and 2017, respectively. Additionally, in economic terms, the scale of annual revenues associated with the recreational use of the forest can be estimated. Previous studies have shown that other factors such as gender, income, and nature of the forest trails influence forest recreation services [57,58]. Fialova et al. [58] expounded on some other factors in their TFE MF Křtiny study. Finally, as the literature indicates, the number of visitors can be an illustrative measure of the recreational use of the forest.

Regarding the literature cited herein, the frequency of visits to TFE MF Křtiny before, during and after the COVID-19 lockdown illustrates the positive significance of forest recreation services' as related to the number of visitors. Even after the suspension of the lockdown (2021), the trend was observed that more visitors went to the studied forest of TFE MF Křtiny, especially during the spring and autumn of 2021, as compared to 2016 and 2017. We conclude that the role of forest recreational services cannot be underestimated in terms of the physical and psychological welfare of the people in a pandemic such as COVID-19 [2,11,12]. Brno city and the surrounding areas and regions in the Czech Republic suffered a severe COVID-19 pandemic situation, with thousands of new infections as well as daily deaths. This resulted in restrictions in movement of people, especially for non-essential workers. Moreover, local tourism and or outside activities were restricted for a majority of the time [19,59,60].

Our study was limited in terms of the factors that influence forest visitors, such as social economic variables. Moreover, another limitation was lack of consideration of important factors such as why an individual would choose a forest to go to, especially regarding the characteristics and quality of the forest. Future studies could focus on the quality of the forests and forest recreation services.

The following conclusions can be drawn from the analysis of the data in this work:

1. The patterns of the visitor number changes are similar throughout the investigated months (March–December) of the years under research (Figure 5).
2. The number of visitors in the lockdown year (2021) was visibly (but not significantly) higher than in the reference years (2016 and 2017) (Table 3, Figure 6), especially in the spring (March, April, May) and autumn (September, October, November) (confirming conclusions from our previous article [19]).
3. The weather conditions in all the years studied did not present limitations (20% participation) to the recreational use of the urban forest; the lockdown year (2021) showed less favourable weather conditions for visiting forests, namely lower mean temperature and higher average rainfall (Table 3), which likely was the reason for the more minor than the expected increment in visitation frequencies during the COVID-19 year. This was undoubtedly accompanied by a desire not to expose oneself to unfavourable weather conditions and to lower infection resistance.
4. This proves the increased need for people to use natural resources (e.g., urban forests) in conditions of a collective health threat and limitations on the freedom of direct interpersonal contacts imposed by the COVID-19 pandemic.

Most Czechs do not follow the rules imposed by the government. Restrictions are not an appropriate solution for the national level. However, since this was an unprecedented situation that no one had experienced before, the measures were in keeping with the times. However, we can learn from this development and use other tools in the future.

**Author Contributions:** Conceptualization, D.B., J.F. and P.K.; methodology, J.F., P.K., J.Ł. and D.B.; software, E.Y. and J.Ł.; validation, B.F.-A., J.B. and J.Ł.; formal analysis, J.Ł.; investigation, J.F. and D.B.; resources, J.Ł.; data curation, E.Y., J.F., J.Ł. and D.B.; writing—original draft preparation, J.F., J.Ł. and D.B.; writing—review and editing, D.B. and B.F.-A.; visualization, J.B.; supervision, P.K. All authors have read and agreed to the published version of the manuscript.

**Funding:** This research received no external funding.

**Data Availability Statement:** Not applicable.

**Acknowledgments:** The authors would like to thank Jacek Łukaszkiewicz for his valuable feedback and assistance with analysis of multivariable metadata.

**Conflicts of Interest:** The authors declare no conflict of interest.

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
