# Peer review of "Urban Forest Recreation and Its Possible Role throughout the COVID-19 Pandemic"

_forests, doi:10.3390/f14061254_

Round 1

Reviewer 1 Report

The paper represents an analysis of the role of the urban forest of the City of Brno during the Covid-19 Pandemic. This type of approach was particularly important and largely addressed in many urban areas to highlight the significance of urban infrastructure in maintaining the health of the population during a shock, such as the one generated by the pandemic.

The Abstract is extensive and comprehensive.

The Introduction presents a complete assessment of the role and importance of investigations on forest areas for human physical and mental health and particularly during the pandemic.  However, since such approaches have been very extensive at the level of many urban areas, the authors could expand the state-of-the-art related to this subject with more bibliographic references. There are many works that have analyzed the green infrastructure (e.g., urban parks, urban forests, urban gardens) as Nature-Based Solutions to support food supply, health and well-being during the COVID-19 Pandemic that are worth mentioning.

The study area is well described in relation to what the study aims to analyze.

2.2. Stages of research and methods used. The difference between the two categories of research questions formulated by the authors is not clear.

Lines 136-139. The connection between the two categories of research methods used and the three stages of the research is not clear.

The Results are clear and well represented visually through graphs made. However, it would have been interesting to add some qualitative aspects to strengthen the recreational role of the forest during the Pandemic. Some relationships of the interviewed people would have been useful to better understand these aspects.

The Discussions discuss the main findings in relation to results obtained in similar works. This section could be updated in relation to the additions that will be made to the Introduction.

The Conclusions present the most important research findings of the paper.

The paper represents an analysis of the role of the urban forest of the City of Brno during the Covid-19 Pandemic.  This type of approach was particularly important and largely addressed in many urban areas to highlight the significance of urban infrastructure in maintaining the health of the population during a shock, such as the one generated by the pandemic.

In my opinion, the work requires a minor revision as it appeared from the suggestions and comments made.

Author Response

The paper represents an analysis of the role of the urban forest of the City of Brno during the Covid-19 Pandemic. This type of approach was particularly important and largely addressed in many urban areas to highlight the significance of urban infrastructure in maintaining the health of the population during a shock, such as the one generated by the pandemic.

The Abstract is extensive and comprehensive.

The Introduction presents a complete assessment of the role and importance of investigations on forest areas for human physical and mental health and particularly during the pandemic.  However, since such approaches have been very extensive at the level of many urban areas, the authors could expand the state-of-the-art related to this subject with more bibliographic references. There are many works that have analyzed the green infrastructure (e.g., urban parks, urban forests, urban gardens) as Nature-Based Solutions to support food supply, health and well-being during the COVID-19 Pandemic that are worth mentioning.

Comment: A few have been added in the discussion section

The study area is well described in relation to what the study aims to analyze.

2.2. Stages of research and methods used. The difference between the two categories of research questions formulated by the authors is not clear.

Comment: Given the importance of Urban forests recreational services during covid, understanding the factors that influence forest visitors very important for policy formulation to improve the services. One focuses on the forest recteational services can be improved while the other on the influencing factors such as weather. Section improved too.

Lines 136-139. The connection between the two categories of research methods used and the three stages of the research is not clear.

Comment: The first explains how we obtain the qualitative data while the second one the quantitive data. The connection with the third one is that the collected quantitative data is assessed.

The Results are clear and well represented visually through graphs made. However, it would have been interesting to add some qualitative aspects to strengthen the recreational role of the forest during the Pandemic. Some relationships of the interviewed people would have been useful to better understand these aspects.

Comment: We focused on data collected by the machine which did not capture relationships of people, however, the qualitative part captures this part and emphasis has been expounded in the discussion too.

The Discussions discuss the main findings in relation to results obtained in similar works. This section could be updated in relation to the additions that will be made to the Introduction.

Comment: Corrected and improved.

The Conclusions present the most important research findings of the paper.

Comment: Thank you

Comments on the Quality of English Language

The paper represents an analysis of the role of the urban forest of the City of Brno during the Covid-19 Pandemic.  This type of approach was particularly important and largely addressed in many urban areas to highlight the significance of urban infrastructure in maintaining the health of the population during a shock, such as the one generated by the pandemic.

In my opinion, the work requires a minor revision as it appeared from the suggestions and comments made.

Comment: Checked by a native speaker as well as the software.

Reviewer 2 Report

Urban Forest of the City of Brno and its Possible Role as an Aid in the Covid-19 Recreation

Thanks for sharing your findings from a mixed-method study on the impact of urban forests on health before and after COCID-19 pandemic. There are some areas that need strengthening. Below I will outline my concerns.  

Title

The title is misleading. I suggest that the authors rethink the title. The term “Covid-19 Recreation” doesn’t make sense.

Abstract

The abstract is long. In the abstract, the problem statement is unclear. Furthermore, the authors need to mention the research methodology in the abstract.

Introduction

Which gaps are revealed by previous research limitations? It is unclear what the current study contributes.

Overall, there are two major conceptual concerns. The first thing to note is that in the introduction the author reviews quickly a few findings from related studies related to the impact of urban forests on human health and wellbeing. This quick overview leads the authors to say that there has been little empirical research evaluating urban forests' impacts during the pandemic. This leads to the second major conceptual problem which is that what is needed here is something that can be added to the literature by including more comprehensive variables to evaluate the impact of urban forests rather than a limited set of indicators.

What about economic changes from 2016-2017 to 2021-2022? What about the rate of population increase in the study area? Or many other variables that may impact the number of visitors to an urban forest. What about the environment/forest quality?

Another important factor that needs to be taken into consideration is the severity of the lockdown in the study area. In some countries, citizens were forbidden from leaving the house even for leisure activities. This must be clear in the paper.

2.2 Stages of research and methods used

Under this subsection, authors have mentioned 2 RQs which is not covered/supported in the earlier part (introduction). They must be in line. What is the main dependent variable here in this study? Is it health? Is it the number of visitors? Or what? Make it clear.

The classification of research methods into two main categories (A. qualitative and quantitative data collection methods, B. analytical methods) seems improper to me. The second group is related to data analysis.

Why did the authors select 2016-2017 as before COVID-19 but not 2017-2018?

Figure 6 does not need to be reported since it repeats the information in Figure 5.

Discussions and conclusions

- Please rewrite this part.      

- The authors did not write well in this section.       

- It's still out of focus and too long.     

- The author must restate the problem statement addressed in the paper; summarize the overall argument or findings, and explain the implications/contributions he made through this article.

Some minor improvements are required.

Author Response

We would really like to thank our opponent for his comments, which were substantive and helped us to realize the errors and problematic parts of the article. We hope that now is the article and the ideas clear.

Thanks for sharing your findings from a mixed-method study on the impact of urban forests on health before and after COCID-19 pandemic. There are some areas that need strengthening. Below I will outline my concerns.

Title

The title is misleading. I suggest that the authors rethink the title. The term “Covid-19 Recreation” doesn’t make sense.

Comment: A new title has been proposed; Urban Forest Recreation and its Possible Role throughout the Covid-19 Pandemic.

Abstract

The abstract is long. In the abstract, the problem statement is unclear. Furthermore, the authors need to mention the research methodology in the abstract.

Comment: Section has been significantly improved aaccording to your suggestions. The problem statement has been added. See section.

Introduction

Which gaps are revealed by previous research limitations? It is unclear what the current study contributes.

Comment: In context of pandemic and lockdown, our research presents interesting patterns of users monthly attendance in the forest (typical and pandemic conditions) in relation to seasons and weather. Also our research show an attempt to determine the time of year when recreational attendance in the forest was particularly dependent on the impact of the pandemic and lockdown - when people visit forest more frequently compare to reference years.

Overall, there are two major conceptual concerns. The first thing to note is that in the introduction the author reviews quickly a few findings from related studies related to the impact of urban forests on human health and wellbeing. This quick overview leads the authors to say that there has been little empirical research evaluating urban forests' impacts during the pandemic. This leads to the second major conceptual problem which is that what is needed here is something that can be added to the literature by including more comprehensive variables to evaluate the impact of urban forests rather than a limited set of indicators.

Comment: We agree with the reviewer - we try to show in the work that the TFE Masaryk Forest case study illustrates, on the one hand, the trend of increasing the tourist attractiveness of urban forests during the Covid-19 pandemic described in the literature, but on the other hand, each case is slightly different due to the amount of traffic tourist. We also show that factors such as weather have some influence on forest attendance, but the need for contact with nature among people affected by lockdown may be greater, even if the weather in a given year is on average worse than in previous years.

What about economic changes from 2016-2017 to 2021-2022? What about the rate of population increase in the study area? Or many other variables that may impact the number of visitors to an urban forest. What about the environment/forest quality?

Comment: Previous research has captured the socioeconomic variables in the study area. However, given the nature of the problem focused on COVID STRESS and the role of forest recreation, this is why we studied both periods to see the variations. Future studies could focus on the quality of the forests, we appriciate it and thank you for the recommendation.

Another important factor that needs to be taken into consideration is the severity of the lockdown in the study area. In some countries, citizens were forbidden from leaving the house even for leisure activities. This must be clear in the paper.

Comment: Yes, this is why this paper is emphasizing the role of forest recreation to help in planning future lockdown but also considering what services could help people get relieved of stress from the lockdown. For the case of the studied area/Czech Republic, forest recreation services such as forest visits were accepted for as long as it was in ones locality.

2.2 Stages of research and methods used

Under this subsection, authors have mentioned 2 RQs which is not covered/supported in the earlier part (introduction). They must be in line. What is the main dependent variable here in this study? Is it health? Is it the number of visitors? Or what? Make it clear.
Comment: In the presented study, the dependent variable is the attendance of users in the forest, which depends on time (season of the year)

The classification of research methods into two main categories (A. qualitative and quantitative data collection methods, B. analytical methods) seems improper to me. The second group is related to data analysis.

Comment: By introducing this division into methods of data collection A. and data analysis B. we tried to clearly show to the reader the course of our research. This division fully reffers to general description of the research presented in earlier parts of the manuscript.

Why did the authors select 2016-2017 as before COVID-19 but not 2017-2018?

Comment: The data from 2018 did not cover the whole year and this is why it was not selected.

Figure 6 does not need to be reported since it repeats the information in Figure 5.

Comment: The figure 6 shows month-by-month visitor numbers differences between 2021 vs 2016 (green) and 2021 vs 2017 (red), so another aspect than figure 5 – we see this graph as really important so we decided to leave it in the article.

Discussions and conclusions

- Please rewrite this part.      

- The authors did not write well in this section.

- It's still out of focus and too long.     

Comment: We have rewritten this part and shorten.

- The author must restate the problem statement addressed in the paper; summarize the overall argument or findings, and explain the implications/contributions he made through this article.

Comment: Thank you for your general but also detailed remarks. We have tried to comply as fully as possible with the indications given. Major or minor corrections have been made in all parts of the work. The purpose of the work has been simplified. We have tried to explain what is new in our study in the context of literature.

Round 2

Reviewer 2 Report

I am happy with some (not all) changes the authors made to the manuscript. However, I still have a number of suggestions to improve the ms.:

1. Introduction. The paper's contribution is still unclear in the introduction.

2. In response to my previous comment on considering other important and effective variables in the study, the authors mentioned that “Future studies could focus on the quality of the forests …”. So why was this significant limitation of the study not mentioned in the paper as one of the study limitations? The authors must acknowledge all limitations of their study.

3. The authors need to clearly mention the severity of the lockdown in the study area (not only the response sheet).

4. Also, we need to clearly mention the study variables in the manuscript (not only in the response sheet in response to my comment). This is one of the most important parts of the paper and the way the authors presented it confuses the readers.

5. The authors’ response to my comment on research methods classification into two main categories is not convincing. This classification is weird. You must correct this explanation. Otherwise, provide justifications with citations why this claim is correct.

6. The reason why the authors did not consider the data in 2017-2018 must be clearly mentioned in the paper so that the readers understand the reason behind it. 

 Moderate editing of English language required.

Author Response

Dear Reviewer,

Thank you very much for your effort and thorough revision of our manuscript. We are really glad that that some of our previous corrections were accepted. Concerning your current remarks we generally agree with you and have corrected and commented herein.

  1. Introduction.The paper's contribution is still unclear in the introduction.

Comments: Following the list, we introduced short extension in the introduction informing about possible contributions of our research. Also the limitations of our research were described more precisely. Added after the aim of the study.

See lines: 115-120, 138-152, 703-713

  1. Inresponse to my previous comment on considering other importantand effective variables in the study, the authors mentioned that “Future studies could focus on the quality of the forests …”. So why was this significant limitation of the study not mentioned in the paper as one of the study limitations? The authors must acknowledge all limitations of their study.

Comments: The limitation has been added at the end of the discussion. See lines 703-713.

  1. Theauthors need to clearly mention the severity of the lockdown in the study area (not only theresponse sheet).

Comments: See lines 138-152 and 722-726

So we hope that our statement from the previous comment sheet is now clear for the article reader too - In context of pandemic and lockdown, our research presents interesting patterns of users monthly attendance in the forest (typical and pandemic conditions) in relation to seasons and weather. Also our research show an attempt to determine the time of year when recreational attendance in the forest was particularly dependent on the impact of the pandemic and lockdown - when people visit forest more frequently compare to reference years.

  1. Also, we needto clearly mention the study variables in the manuscript (not only in the response sheet in response to my comment). This is one of the most important parts of the paper and the way the authors presented it confuses the readers.

Comments: See lines 370-372

  1. Theauthors’ response to my comment on research methods classification into two main categories is not convincing.This classification is weird. You must correct this explanation. Otherwise, provide justifications with citations why this claim is correct.

Comments: We deleted this part as it was really not necessary to have those statement in the article.

  1. Thereason why the authors did not consider the data in 2017-2018 must be clearly mentioned in the paper so that the readers understand the reason behind it. 

Comments: See lines 295-297 and 698-702